# Prevalence and associated risk factors of HIV infections in a representative transgender and non-binary population in Flanders and Brussels (Belgium): Protocol for a community-based, cross-sectional study using time-location sampling

**Niels De Brier**[1,2⍟]*, **Judith Van Schuylenbergh**[3⍟], **Hans Van Remoortel**[1,2], **Dorien Van den Bossche**[4], **Steffen Fieuws**[2,5], **Geert Molenberghs**[5], **Emmy De Buck**[1,2], **Guy T'Sjoen**[3,6], **Veerle Compernolle**[7,8], **Tom Platteau**[4], **Joz Motmans**[3]

**1** Centre for Evidence-Based Practice, Belgian Red Cross, Mechelen, Belgium, **2** Department of Public Health and Primary Care, Leuven Institute for Healthcare Policy, KU Leuven, Leuven, Belgium, **3** Center for Sexology and Gender, Ghent University Hospital, Ghent, Belgium, **4** Department of Clinical Sciences, Institute of Tropical Medicine, Antwerp, Belgium, **5** Interuniversity Institute for Biostatistics and Statistical Bioinformatics (I-BioStat), University of Leuven and Hasselt University, Leuven, Belgium, **6** Department of Endocrinology, Ghent University Hospital, Ghent, Belgium, **7** Blood Service, Belgian Red Cross, Mechelen, Belgium, **8** Faculty of Medicine and Health Sciences, University of Ghent, Ghent, Belgium

⍟ These authors contributed equally to this work.
* niels.debrier@cebap.org

# Abstract

## Introduction

HIV prevalence and sexual risk have been estimated very high for transgender people. However, the limited sampling and data collection methods used in current research on transgender people potentially led to overrepresentation and generalisation of people at risk for HIV. Current HIV prevalence estimates in transgender populations are generalised from studies mainly focusing on transgender women engaging in sex work. Moreover, studies focusing on non-binary people, who identify with a broad range of identities beyond the traditional male and female gender identities, are scarce.

## Objectives

To estimate the HIV prevalence rate in the Flemish and Brussels (Belgium) transgender population, including transgender women, transgender men and non-binary people, and to identify the associated risk factors.

## Methods

In this community-based cross-sectional study, self-identified transgender and non-binary (TGNB) people will be recruited through a two-stage time-location sampling approach. First, community settings in which TGNB people gather will be mapped to develop an accurate

**Funding:** This work was made possible through funding from the Foundation for Scientific Research of the Belgian Red Cross (Mechelen, Belgium), the Institute for the Equality of Women and Men (Brussels, Belgium), University Fund Ghent University (Ghent, Belgium) and Gilead Sciences (Diegem, Belgium; Grant number: BE-2021-0049). The funding body had and will not have a role in study design, data collection and analysis, decision to publish, or preparation of the manuscript.

**Competing interests:** The authors have declared that no competing interests exist.

**Abbreviations:** aOR, adjusted odds ratio; AMAB, assigned male at birth; AFAB, assigned female at birth; CBPR, community-based participatory research; ELISA, enzyme-linked immunosorbent assay; ICC, intra-cluster correlation coefficient; LGBT+, lesbian, gay, bisexual, transgender and other; MSM, men who have sex with men; OR, odds ratio; STD, sexually transmitted disease; TGNB, transgender and non-binary; TLS, time-location sampling; WHO, World Health Organization.

sampling frame. Secondly, a multistage sampling design is applied involving a stratification based on setting type (healthcare facilities vs outreach events), a selection of clusters by systematic sampling and a simple random selection of TGNB people within each cluster. Participants will complete an electronic self-reported survey to measure sociological, sexual and drug-using behaviors (risk factors) and oral fluid aliquots will be collected and tested for HIV antibodies. Logistic regression models will be used to evaluate risk factors independently associated with HIV infection. The presented study is registered at ClinicalTrials.gov (NCT04930614).

## Discussion

This study will be the first to investigate the HIV prevalence rates and associated risk behaviors in an accurate representation of the TGNB population in a Western European country. The findings will globally serve as a knowledge base for identifying subgroups at risk for becoming infected with HIV within TGNB people and to set up targeted prevention programs.

## Introduction

Globally, 0.7% of all adults aged 15 to 49 years are living with HIV [1]. The prevalence of HIV infection is estimated at 0.4% and 0.17% in Europe and Belgium, respectively [2]. The World Health Organisation (WHO) has listed five key populations at risk for HIV: men who have sex with men (MSM), sex workers, people in prisons or closed settings, people who inject drugs and transgender people [3]. Indeed, over the past 15 years, HIV prevalence has been estimated high for transgender people.

Transgender people are people who do not or to a lesser extent identify with their birth-assigned gender: transgender women are people who were assigned male at birth (AMAB) but identify on the female spectrum; transgender men are people who were assigned female at birth (AFAB) but identify on the male spectrum. Those who do not identify as binary trans, but who identify between, outside and beyond the gender binary are referred to in this article as non-binary people.

A global HIV prevalence rate of 19.1% has been calculated among transgender people, based on 39 studies in 15 countries [4]. A meta-analysis based on 29 US-based studies estimated a self-reported HIV prevalence rate of 11.8% and laboratory-confirmed HIV prevalence rate of 27.7% [5]. A recent update of this meta-analysis, based on 88 studies published between 2006 and 2018 calculated an overall HIV prevalence of 13.7% and an HIV prevalence of 18.8% among transgender women [6]. Furthermore, this study calculated a self-reported history of sexually transmitted disease (STD) diagnosis rate of 21.5%.

Most studies on HIV identify high rates of sexual risk behaviors among transgender people, such as unprotected intercourse, having multiple partners and having sex while drunk or high [4]. Engagement in sex work is also frequently reported, for transgender women as well as for transgender men. The meta-analysis of Becasen et al [6] reports that 31% of all transgender people had a history of sex work: 37.9% of all transgender women and 13.1% of all transgender men. The engagement of transgender people in sex work is frequently attributed to discrimination on the job market, leading to economic marginalisation and forcing transgender people in sex work to survive [7]. Discrimination and rejection have been significantly associated

with sexual risk behaviour [8]. Stigma and discrimination may lead to lower self-esteem, mental health issues and the need for acceptance and affirmation from others, which may lead to lower self-efficacy to negotiate safe sex [9]. As such, transgender people experience multiple and overlapping vulnerabilities which have an influence on their health.

However, although HIV prevalence is considered high among transgender populations, prevalence rates also tend to vary widely. A recent systematic review shows that prevalence rates range between 0% and 49.6% across studies [10]. Hence several researchers have been critical towards current HIV research in transgender people more specifically regarding the samples used as well as the lack of accounting for intragroup variation, and the lack of inclusion of community perspectives [11, 12].

## Sampling issues

Oversampling of transgender people at high risk for HIV appears to be common: a lot of studies are focused on transgender women or (female) transgender sex workers [4, 10–12]. For example, the recent systematic review of van Gerwen et al. [10] included 25 studies. All of these studies included transgender women but only 9 included data on transgender men, often in small samples. A lot of transgender HIV research has been conducted in sexual health clinics and community centres in large urban areas, resulting in samples of transgender people at high risk for HIV [4, 10–12].

Obtaining a generalizable sample of transgender people is difficult due to a lack of reliable up-to-date information regarding the number and the proportion of TGNB people in the general population [13]. Hence, transgender health research is based on either clinical samples of transgender people attending specialized transgender healthcare clinics, or on convenience community samples. Clinical samples are limited since a significant proportion of the transgender community either has no access to transgender healthcare or does not want or need medical interventions. Findings from the recent European Union Agency for Fundamental Rights (FRA) study show that up to 65% of all Belgian transgender respondents and 72% of all European transgender respondents had not undergone any medical intervention to change their body in order to match their gender identity [14]. Studies using broad convenience community samples usually recruit via LGBT+ (lesbian, gay, bisexual, transgender and other) or specific transgender organisations, whereas not all transgender people are connected to such organisations–certainly not years after transitioning. The FRA study for instance shows that 72% of all Belgian transgender respondents and 76% of all European transgender respondents are not actively involved in an LGBT+ organisation [14]. Some subgroups of the transgender community remain largely invisible, for instance those who have not yet come out as being transgender, those who live 'stealth' or people who are homeless [15].

It becomes clear that as long as gender identity is not taken into account in general population research, obtaining a generalizable sample of transgender people is nearly impossible [16–18]. Some recent studies have attempted to obtain more representative samples of transgender people. A large online sample of US-based transgender people shows for instance a self-reported HIV rate of 2.2%, which is much lower than estimates cited in other studies using self-reported data [19]. The recent meta-analysis of Becasen et al. [6] indeed shows relatively lower estimates than previous meta-analyses, which does not indicate a decrease in HIV infections but more likely reflects research slowly beginning to include transgender people from locations and backgrounds that might be more representative of the overall transgender population [6].

## Intra-group variation

Although HIV prevalence has been estimated relatively high for transgender women (0–49.6%), HIV prevalence is estimated much lower in transgender men, ranging between 0 and 8.3% [5, 6, 10], however few studies include or specifically focus on this subpopulation. Having (cisgender) female sexual partners and lower participation in sexual risk behavior are suggested explanations for this lower HIV prevalence, but more research is warranted [6]. Most current HIV research remains highly heteronormative, focusing on transgender women engaging in receptive anal intercourse with cisgender men. Having sex with cisgender men, regardless of the gender identity of the person itself, has indeed been shown to be a high risk factor for transgender people [19]. However, transgender women with female partners as well as transgender men with male partners and transgender people with transgender partners have been largely ignored in current HIV research. A Canadian study reported that within their sample of transgender men (n = 227) only 34.3% were identified as heterosexual, and 31% had a (cisgender or transgender) male partner during the past year [20]. Of these transgender men who are gay, bisexual or MSM (GB-MSM), 9.2% had high risk sexual intercourse, 14.1% had more than 5 sex partners during the past year and 15.8% had ever been engaged in sex work. Reviews of MacCarthy et al. [12], Van Schuylenbergh et al. [11] and Van Gerwen et al. [10] further show that current sociological and serological research also remains binary and cisnormative. Research including non-binary transgender people, who identify with a broad range of identities that lay between or beyond the traditional male and female gender identities, is scarce to non-existent. Furthermore, prevalence rates seem to be higher for transgender women of color: the US-based systematic review of Herbst et al. [5] calculated a HIV prevalence of 56.3% for black, 16.1% for Hispanic and 16.7% for white transgender people. Compared to white transgender women, black and Hispanic transgender women seem to be more involved in sex work and report more unprotected receptive anal intercourse [21].

In summary, intra-group variation within the transgender population is clear, but often overlooked, and perspectives that account for the intersection of different identities still lack in current transgender HIV research.

## Community perspectives

In the past, research involving transgender people has been pathologizing, stigmatizing and even transphobic [22, 23]. Researchers have for instance systematically misgendered transgender people, regarded 'transgender' as a category of sexuality and conflated lesbian, gay, bisexual and transgender populations [22]. Specifically for HIV research, categorising transgender women as MSM was, and often still is, a common practice [24]. This has led some transgender people to distrust researchers investigating transgender health or identities [24]. Furthermore, lack of transparency in communication by researchers, feeling used for research and not hearing about findings from research they participated in are important issues raised by transgender people, often leading to research mistrust [15, 24]. This is why 'community involvement' is considered an important aspect of ethical research in transgender populations [22, 24–27]. Furthermore, researchers stress a need for multiple modalities and methods to reach different subsets of the transgender community, preferably using peer recruitment techniques and ensuring transgender participation in advisory groups, to make sure hard-to-reach groups are included too [15, 24].

Overgeneralization of findings to the transgender population as a whole still persists, and should be critically investigated. Studies have been limited to specific regional and contextual settings, such as large urban centres in North-America (San Francisco, Los Angeles), although the social as well as the legal context for transgender people greatly differs between countries

and regions. Unpublished data from a large European study on transgender health suggests that self-reported HIV prevalence in European transgender people is estimated lower than the prevalence rates often cited in the mainly US-based literature, around 1.6% [28]. However, European peer-reviewed articles that provide actual and current information on HIV prevalence and sexual health in a representative sample of transgender people are not available.

With this research protocol we present a study investigating HIV prevalence in the heterogenous transgender population in Flanders and Brussels (Belgium). In contrast to previous transgender HIV research, this study specifically aims to include non-binary transgender people. This is why the term 'transgender and non-binary (TGNB) people' is further used to describe the study population.

## Objectives

This protocol describes the setup of a study aiming to provide a first HIV prevalence estimate for a representative sample of TGNB people residing in a Western European country by combining state-of-the-art epidemiological and sociological methods. By applying a Community-Based Participatory Research (CBPR) and Time-Location Sampling (TLS) approach, this study will allow us for the first time (i) to qualitatively build a sociological map of community settings in which TGNB people congregate for building a sampling frame, (ii) to determine the prevalence of HIV infections in the Flemish and Brussels TGNB population and (iii) to identify associated (sexual) risk factors. The study hypothesis is threefold: (i) social networks and settings where TGNB people may congregate exist, (ii) TGNB people have relatively high rates of HIV infection but differences in HIV prevalence across this heterogeneous population may exist and (iii) the HIV prevalence is mainly determined by sexual risk behavior.

## Study design

We will conduct a community-based, cross-sectional study in which HIV prevalence will be estimated and associated with sociological, sexual and drug-using behavioral data. To obtain a representative sample of TGNB people, including hidden groups such as those who do not need medical assistance, we opted for Time Location Sampling (TLS), which takes advantage of the fact that some hard-to-reach groups tend to gather at certain (online) settings [29]. In more concrete terms, TLS approximates probability sampling by first mapping the universe of settings where TGNB people may gather and then applying a two-stage random sampling strategy. The latter involves the random selection of settings at specific time periods followed by random participant recruitment within each setting [29–31]. By applying the TLS approach, this study consists of two main parts: (i) a formative qualitative study to map social and medical settings frequented by TGNB people, which will include physical as well as digital community settings in Flanders and Brussels (Belgium) and (ii) a cross-sectional analysis to estimate HIV prevalence among Flemish TGNB people and identify associated risk factors.

After the development of the sampling frame, TGNB people will be recruited in a wide range of settings using a two-stage TLS frame for obtaining a representative sample. In this TGNB population, HIV will be screened through oral fluid samples and, at the same time, sociological, sexual and drug-using behaviors will be surveyed to reveal the association between HIV prevalence and these risk-taking behaviors. In case of a reactive oral fluid result, confirmation on a blood sample is required to allow accurate estimation of HIV prevalence. After completion of the study, we plan and report this observational study in accordance with the STROBE checklist (Strengthening the Reporting of OBservational studies in Epidemiology). Although not an interventional study, we used the relevant items of the SPIRIT checklist

to report this protocol paper (S1 Checklist) and an SPIRIT overview of the schedule of assessments can be found in Fig 1.

## Methods

### Study area

We will conduct a broad survey in two out of three regions of Belgium (Flanders and Brussels). Flanders is the Dutch-speaking northern region of Belgium and counts approximately 6.6

| | Study period | |
|---|---|---|
| | **Enrolment** | **Close-out** |
| **TIMEPOINT** | *0* | *Two weeks* |
| **ENROLMENT:** | | |
| **Eligibility screen** | X | |
| **Informed consent** | X | |
| **ASSESSMENTS:** | | |
| *Socio-demographic variables* | X | |
| *Medical history* | X | |
| *Risk behaviors* | X | |
| *Additional socio-demographic variables\** | | X |
| *Transgender specific variables\** | | X |
| *Additional risk behaviors\** | | X |
| *HIV diagnostic orientation test* | X | |

**Fig 1. Schedule of enrolment and assessments.** *Because of recruitment taking place in a variety of non-research settings, a relatively short questionnaire needs to be completed at enrolment, and a follow-up questionnaire with additional questions will be sent to the participants by email (cfr. PART II).

million inhabitants in 2020. It is characterized by high land take, with many urban and rural centres. The Brussels-Capital Region is located in the central part of Belgium comprising 19 municipalities, including the City of Brussels, which is the capital of Belgium. This region has a population of around 1.2 million [32]. In total, Flanders and Brussels cover about 68% of the total Belgian inhabitants. Furthermore, large differences between the regions exist in terms of language, policies, transgender healthcare and services and the organization and structure of the TGNB community. Due to limited resources and difficulties in penetrating into the TGNB community which is to date not well sociologically mapped, TGNB people residing the French-speaking southern part of Belgium, the Walloon region, are excluded in this study. We believe that this focus will benefit the representativeness of our sample.

## Community involvement: Advisory board and peer data collectors

Throughout the study, CBPR methods will be used to ensure community involvement. CBPR entails participation of the target community throughout all phases of the research process, and is often used in research involving hard-to-reach populations. CBPR models offer mutually beneficial collaborative partnerships between those who fund, sponsor and implement research and the groups, individuals and communities affected by it [25]. Using CBPR methods, TGNB people become co-producers of knowledge [22]. In concrete terms, we will first set up a community advisory board consisting of a diverse range of TGNB people, which is strongly advised for research involving transgender people [25]. At the time of writing, the community advisory board was already composed by the researchers and consisted of 15 TGNB persons (including transgender women, transgender men, queer and non-binary persons). This advisory board will be consulted throughout several phases of the study. In first instance, the acceptability of the study within the TGNB community was already estimated by the community advisory board and we investigated how TGNB people want to be involved in studies investigating HIV in TGNB people. As a result, the community advisory board will also co-create the branding of the study, validate the community map resulting from the in-depth interviews, provide direct input on the content and comprehensibility of the questionnaires of the study, on the recruitment process and on the communication strategy of the published results. The community advisory board will not take part in outcome assessment and data analyses which will be executed by researchers who will not be involved in data collection. Consulting the community advisory board can take place online as well as offline on a regular basis, depending on the preferences of the community advisory board members, ensuring the accessibility of the setting.

Furthermore, 60% of the main research team consists of TGNB community members, and in addition three peer data collectors will support the researchers during data collection at community events and meetings during the study. Having TGNB peers involved as part of the research team appears to be a strong indicator for participants' willingness to participate in transgender health studies [15, 22, 24]. Furthermore, peer data collectors experience fewer hurdles to overcome mistrust between researchers and hard-to-reach communities. However, the participation of TGNB people in the research team can also create a number of challenges, such as potential bias, blind spots and interpersonal factors [25]. Hence, peer data collectors require training to address challenges related to peer-to-peer research interactions, as well as to ensure all participants are treated with respect and culturally competent language is used.

## PART I: Formative community mapping study

To set up the sampling framework, a formative study is first carried out aimed at mapping community settings in which TGNB people congregate, using qualitative and ethnographic

research methods. The main research questions of this formative study are: (i) how is the TGNB community in Flanders and Brussels structured, and (ii) in what settings can we recruit TGNB people in order to obtain a representative sample?

**Sampling method and data collection tools.** We use part of the systematic approach of the PLACE method (Prioritizing Local AIDS Control Efforts) to map the TGNB community [33]. The identified area to map is Flanders, the Dutch-speaking part of Belgium, and Brussels. We will use a snowball sampling method starting from a diverse convenience sample of key informants (seeds) in terms of gender identity, ethnicity, age and sexual orientation and will keep on collecting data until saturation occurs (i.e., no new settings result out of the interviews). Additionally, we will use tools such as the Transgender Healthcare Map of the Transgender Information Point, the Flemish expertise centre for the transgender theme, to map transgender healthcare services and peer online and offline groups. Qualitative and ethnographic methods will be used to map community settings, using participant observation and informal conversations at different settings, as well as semi-structured in-depth interviews with key informants to collect data. Semi-structured interviews will be recorded (given consent of the key informant), transcribed and coded using NVivo Software for qualitative analysis. For participant observation and informal conversations, field notes will be registered and coded in an analogous way. All data will be pseudonymized. To control whether saturation has been accurately reached, study participants in PART II of the study will be asked for what activities, healthcare services and/or digital spaces they are aware of. If new community settings are identified, they will be included in the quarterly sampling frame (cfr. PART II).

**Study population.** Key informants include TGNB community members as well as other people affiliated with the community, such as healthcare workers, social workers or people affiliated with LGBT+ organisations.

**Study parameters.** Key informants will be asked what physical places, venues and/or events they know that TGNB people frequent, what kind of activities or services are organised there and how many and what kind of visitors come to these settings. Similarly, key informants will be asked what digital spaces for TGNB people they are aware of, what kind of topics, messages or interactions are posted in these spaces and how many and what kind of people are part of these digital spaces. During in-depth interviews, visual techniques will be used to further explore the structure of the TGNB community: a visual map of the TGNB community based on the data already collected will be shown to each new key informant and completed with the information gathered during the conversation, until saturation occurs. Additional questions gauging the acceptability of the study design include questions on community involvement in research and barriers and facilitators for participation in research.

**Setting eligibility.** Settings eligible for consideration for recruitment in the main study are defined as public, private or online digital settings attended by TGNB people for any purposes, including (but not limited to) medical, mental health and social services, pride events, bars, discussion groups, social media groups, online fora,. . . All settings where TGNB people congregate will be included. Hence, we specifically aim at mapping digital community spaces too, as some TGNB people are unlikely to be reached using in-person methods, simply because they are not enrolled in transgender healthcare, are not active in LGBT+ or transgender community groups and do not attend LGBT+ or TGNB oriented events. As Vincent [22] notes, *"the conceptualisation of 'community space' as a location where individuals who belong or are connected to a particular group may meet, bond, resist oppression, share resources or find a sense of connection, extends beyond physical geography. For many [transgender people] digital community spaces are highly significant."*

**Development of the sampling frame.** The data obtained through this formative study will result in a list of settings and their characteristics in terms of activities and visitor profiles.

All community and healthcare settings will then be contacted to verify the obtained information, to estimate the attendance data of TGNB people and to ask for their willingness to facilitate the recruitment of TGNB people in an HIV study. We will also reach out to online group administrators and ask permission to potentially use these spaces for participant recruitment [22]. Study recruitment without permission and contextualisation can come across as highly intrusive to TGNB online community group members.

## PART II: Cross-sectional community-based study on HIV prevalence and associated risk factors among TGNB people in Flanders and Brussels

The list of candidate venues established in Part I will here be used as the sampling frame. The sampling frame will first be divided into two strata (healthcare settings and physical and digital outreach events) from which a two-stage cluster sample will be selected to study the HIV prevalence and to identify associated risk factors in a representative TGNB study population. Stratification ensures that TGNB people in each setting type receive proper representation within the sample. At the first level of sampling, clusters will randomly be selected from the developed sampling frame using systematic sampling. Clusters are defined as settings which are attended by TGNB people with their associated days and time periods. The second level of sampling will include the random selection of TGNB people from each cluster [31, 34].

**Study population with eligibility criteria.** Population studies in the Netherlands [35] and Flanders [36] indicate that respectively 0.9% and 0.6% of their populations could be classified as 'genderincongruent', thus as identifying more with the opposite gender than the gender which was assigned at birth. Furthermore, respectively 3.9% and 2% of their populations could be classified as 'genderambivalent' or identifying with both or neither binary genders. Extrapolated to a Flemish population of 6.6 million citizens [32], this amounts to more than 170.000 TGNB people living in Flanders. For this study, self-identified TGNB people, aged 18 or older will be eligible to participate once, given that they consent to fill in the questionnaire as well as donate an oral fluid sample. Transgender women and transgender men, as well as non-binary people are included in this study. Transvestite or crossdressing people who identify as transgender are also included. Previous research demonstrated that cross-dresser and transvestite are indeed terms (a minority of) transgender people use to describe their gender identity in Belgium [37]. We are aware that, although non-binary identities are considered as part of the transgender umbrella, not all non-binary people identify with the term 'transgender'. As such we use 'identifying as TGNB' as the inclusion criterium regarding gender identity. Cisgender people and participants who already participated in the study are excluded. There are no restrictions for nationality provided that the participants have an understanding of the multilingual survey (cfr. infra). Finally, eligible TGNB people should reside in Flanders or Brussels at the time of data collection.

**Sample size.** Recent large online surveys indicated that the self-reported HIV prevalence rate in a large sample of transgender people in Europe and the US was 1.6 [15] and 2.2% [24], respectively. Since it was estimated that 30% of HIV-infected people in the European Union have not been diagnosed [38, 39], the sample size was calculated to detect an HIV prevalence rate of 3% with a precision of 1.5%. In total, 560 TGNB people need to be recruited for participation such that the expected length of the confidence interval amounts 0.03 (± 0.015) when the proportion of HIV infection is 0.03 and the level of confidence is 95% based on power test for binomial responses (RStudio, Version 3.6.1, RStudio Inc. Boston, MA, USA; Package 'exactci').

Because TGNB people recruited from the same cluster may share characteristics such as risk behaviors or demographics, their outcomes might be correlated. It is common to adjust

for this correlation by multiplying the sample size required for a simple random sample by the "Design Effect," also known as the "Variance Inflation Factor". The design effect is a function of the number of individuals per cluster, m, and the intra-cluster correlation coefficient (ICC), describing the proportion of the total variation in outcomes that is due to variation between clusters.

$$Design\ Effect = 1 + (m - 1)ICC$$

We rely on previously published ICCs for HIV outcome for the sample size calculations. However, ICCs depend on, amongst others, study's outcome, design and population but, to date, there are no published ICCs that apply to HIV in TGNB people. We are aware of one study that used TLS in detecting HIV infection in MSM and reported ICCs (kappa-like estimator) ranging from 0.03 to 0.16 (0.08 ± 0.06) [40]. Since the mean cluster size is currently unknown, we can only estimate the design effect in advance. The cluster sizes will be fixed at multiples of ten (cfr. infra) and we expect that the proportion of small clusters will be relatively large. The average number of TGNB people sampled per cluster will hence be close to ten but the impact of the large events on the average cluster size remains to be revealed in this study. Although an average cluster size of 12 to 14 TGNB persons would be realistic based on theoretical simulations, we prefer to be conservative and to inflate the cluster size in the sample size calculations. When we for example assume on average 18 TGNB people per cluster, the design effect would be 2.36 [= (1+(18–1)*0.08] (varying between 1.51 and 3.72). Analysis of the data of the TLS involving MSM showed that the design effects may be large and that especially variation in the weights may increase the design effects [40]. To our knowledge, the importance of the design effects in the analysis of TLS data has rarely been considered. Note that the calculation of the impact of the design on the required sample size is an approximation, not only because the mean cluster size and the variability in the cluster sizes are unknown, but also since the sampling weights (cfr. infra) were not considered.

At this stage, when assuming a design effect of 2.36, 1322 TGNB people should be recruited to determine the HIV prevalence.

$$n = Design\ Effect * 560 = 2.36 * 560 = 1322$$

Finally, to assess the associated risk factors of HIV infection among TGNB people, we determined the minimum sample size for comparing two proportions in the cross-sectional study using following formula:

$$n_{Fleiss} = Design\ Effect * \frac{\left[Z_\alpha\sqrt{(r+1)p(1-p)} + Z_\beta\sqrt{rp_0(1-p_0) + p_1(1-p_1)}\right]^2}{r(p_0 - p_1)^2}$$

$$p = \frac{p_0 + rp_1}{r + 1}$$

Where n is the sample size in the smallest group, Z is the statistic corresponding to 95% level of confidence (1.96), $Z_\beta$ is the statistic corresponding to the 80% power level, $p_0$ is the proportion of HIV infection in population with risk behavior, $p_1$ is the proportion of HIV infection with no risk behavior, r is the ratio of sample sizes in the two populations ($p_1/p_0$). Since the risk factor will be assessed at the level of the individual TGNB person, the use of the design effect for clustering is a conservative approach here. Indeed, the design effect will be smaller than with the prevalence estimate since the risk factor of interest varies within a cluster.

As an example, 812 TGNB people are needed to have 80% power to detect a risk factor associated with HIV infection when assuming that the proportion of HIV is 0.06 in the population

risk behavior ($p_0$) and 0.005 in the population with no risk behavior ($p_1$) and the ratio of sample sizes (r) amounts to three (Table 1). Other, more conservative, estimations are also outlined in Table 1 and, in any case, the number of TGNB people needed for detecting differences in risk behavior is well below the sample size needed for estimating the HIV prevalence.

However, since little information on the elements determining the design effects (ICCs, mean cluster size, variability in cluster sizes and sampling weights) is currently available (cfr. supra) and the proportion estimates are relatively uncertain, an interim analysis for sample size recalculation will be performed after the data of the TGNB people recruited at the first two quartiles (cfr. infra) are available. When a larger sample is required to guarantee the desired level of precision (being half of the expected prevalence rate), more clusters will be selected in the final sampling calendar. When a lower sample would suffice, the sample size remains as planned [41]. It is important to note that maximal 1500 TGNB people can be recruited within the foreseen budget.

**Sampling method.**    In the design of the sampling scheme both the coverage of the TGNB people in Flanders and Brussels and the logistic feasibility of the fieldwork are important concerns. In first instance, the sampling frame will be **stratified** based on the type of the setting where TGNB people may gather. With stratification TGNB people can be recruited at

i.  healthcare facilities providing TGNB specific care such as gender affirming hormone therapy, surgery, and/or psychological support, and

ii.  physical and digital outreach events (e.g. LGBT+ bars, prides, discussion groups, social activities of LGBT+ organisations, TGNB specific digital forum groups, social media groups but also digital TGNB networks of influencers).

Although it was recently estimated that 35% of the TGNB people did undergo any medical intervention to change their body [14], this estimate is uncertain and has to be confirmed in this study. Therefore, at this stage, 50% of TGNB participants in this study will be recruited from healthcare settings and 50% from physical and digital outreach events. By allocating the same number of respondents to each stratum, the precision of the prevalence estimates at the level of the different strata will be about equal. It is of note that a potential difference in selection probability does not impact on the representativity of the study sample. Indeed, when combining the strata results to the total population level, a certain degree of precision will be lost, but a valid prevalence estimate can be obtained by weighting each setting type with weights inversely proportional to the selection probability. Importantly, strata used in this study will overlap, because some individuals will e.g. receive TGNB specific medical care *and* regularly attend LGBT+ bars. These TGNB people will have a higher chance of being selected than others. To overcome this limitation, participants will be asked for their frequency of attendance between and within different strata to estimate overlap. (Individual) post-stratification weights will hence be used in the data-analysis to calculate the setting and overall estimates (cfr. infra).

**Table 1. The minimum sample size for comparing two proportions assuming a two-sided significance level of 95%, a power of 80% and a ratio of sample sizes of three.**

| Proportion of HIV in TGNB population with risk behavior ($p_0$) | Proportion of HIV in TGNB population with no risk behavior ($p_1$) | Total sample size |
|---|---|---|
| 0.060 | 0.005 | 812 |
| 0.045 | 0.005 | 1213 |
| 0.035 | 0.001 | 1150 |
| 0.060 | 0.010 | 1110 |

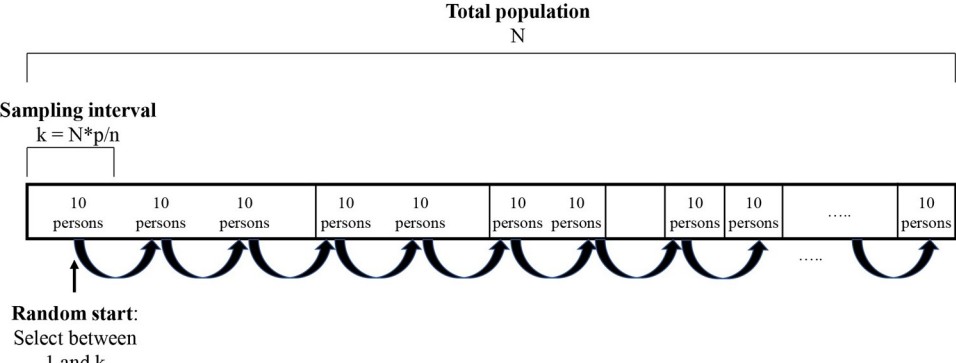

**Fig 2. Sampling scheme for the selection of clusters in which each cluster is represented by a segment proportional to its expected number of attendees with k the sampling interval, p the number of subjects per sample group and n the required sample size.** The sum of the segments represents the total population (N).

For each stratum, a **two-stage TLS** will then be used to recruit the required number of study participants. Consultation periods are used as clusters (primary sampling units) in the stratum of the healthcare settings while the clusters in the outreach event stratum are defined based on the identified physical and digital venues at well-defined time periods. To determine in which clusters the TGNB people will be selected, they will first be ordered by size from large to small based on expected number of attendees and, at the same time, expanded on the list proportional to their size. For determining the number of attendees in healthcare facilities, the exact number of patients attending consultation periods at specific time slots will be provided by the counsellors or the hospital data managers for the upcoming three months. Regarding the outreach events (e.g. pride events, parties or gatherings in bars, discussion groups, . . .), the organizers will be contacted for collecting data on the expected number of TGNB persons. In social media groups or online fora, the researchers will have access to the full list of members after approval by the group administrators. A stepwise systematic selection procedure is then used to select clusters based on a random start and an interval equal to the total population size divided by the required sample size (Fig 2) [42, 43]. As a consequence, the selection chance of each cluster will be proportional to its expected number of attendees and, hence, one ensures that large, medium, and small clusters are all present. More in particular, large events which attract a large number of eligible respondents are certainly selected and the division over smaller and larger clusters is about equal to the real distribution. Hence, some large clusters will be selected more than once because their size based on expected number of attendees is a multiple of the step size by which the weighted systematic sample will be taken (Fig 2). The number of participants in each cluster should be at least ten in order to keep the fieldwork manageable. Hence, it was decided to work with multiples of ten within each cluster. To make this procedure more transparent a graphical representation (Fig 2) was given.

This selection of clusters will be made for each quarter during one year. It was deliberately chosen to not perform the selection procedure for the whole year at once since the characteristics of the clusters (e.g. availability and expected number of attendees) might change during the time the study is conducted, especially given the restrictions related to the current COVID-19 pandemic. Prior to each selection step, the sampling frame will be updated by contacting the key informants and cluster owners and organizers (cfr. Part I). To have representativity over time, the required sample size should be spread as equally as possible over the different quarters.

If a selected cluster was unavailable or declined to facilitate the sampling procedure, we will document this and randomly select the previous or next most similar cluster on the systematic sampling list. By a-priori matching the replacement clusters based on size (i.e., expected number of attendees) and type of setting (e.g. LGBT+ bar will be replaced by another LGBT+ bar), we aim to avoid systematic trends in drop-out. In the specific case that a cluster will temporarily be unavailable due to restrictions related to COVID-19 pandemic (e.g. mass gatherings), the cluster will be visited on a later timepoint when possible. The selection procedure will be done by an independent researcher (NDB) who will not be involved in data collection and outcome assessment using RStudio.

In a second step, the participants present at the cluster at the time of data collection (secondary sampling units) will be selected based on a sequence of random numbers that will be generated by NDB. If the target of participants (at least ten people) could not be reached, multiple visits will be made to the same setting until the desired number of participants is achieved. When not successful, it is equally possible to select the previous or next most similar cluster on the systematic sampling list for recruiting the remaining number of eligible TGNB people (cfr. supra). In the case that TGNB people refuse to participate, the reason(s) behind their refusal and their gender and age will be documented (if possible). Additional sampling will be performed at the cluster based on the sequence of random numbers to reach the required number of respondents.

The feasibility of the sampling method will be evaluated and, if deemed necessary, adjusted at two phases of the project: (i) after completion of the sampling frame (cfr. Part I) and (ii) after analysis of the data of the TGNB people recruited at the first ten clusters.

**Description of variables.** *Gender identity*. To measure gender identity we will ask respondents to categorize themselves in one of the following categories, in order to be able to interpret the results of the study: (i) (transgender) man, (ii) (transgender) woman, (iii) non-binary or genderqueer or (iv) transvestite or cross-dresser. Whereas the two-step method—asking gender as well as sex assigned at birth—has long been advised for measuring gender identity, we will only ask for sex assigned at birth to respondents identifying as non-binary/genderqueer or transvestite/cross-dresser. Asking transgender people what sex they were assigned at birth should be avoided when not relevant, as such this is deemed unnecessary for transgender men or transgender women, but is important information for people identifying as non-binary, genderqueer, transvestite or crossdresser.

*Other socio-demographic variables*. Besides the demographic characteristics (age, city of residence, country of birth, country of birth of both parents, ethnicity, marital status, living situation, education, financial situation), variables specific for TGNB people will be surveyed (active in community, attendance to events, activities, services and digital spaces they are aware of).

*Medical history*. Past medical treatments such as blood transfusion or organ transplant may put the participant at risk of HIV infection and will be questioned as confounding factors along with access to transgender-related healthcare (gender affirming hormone therapy and surgery and/or psychosocial assistance) and sexual health history (history of STDs and HIV testing history).

*Risk behaviors*. The sexual and intravenous drug-using risk factors will be surveyed based on the time-based deferral criteria of the donor health questionnaire of Belgian Red Cross-Flanders. The donor health questionnaire primarily aims to identify risk behavior for potential transfusion-transmissible infections. We will here implement these questions in the survey for assessing the risk factors for HIV infection: sexual risk contacts (e.g. new partners, group sex, multiple partners or transactional sex) and the use of needles for injecting drugs, hormones or

silicone. In addition, to fully understand these risk-taking behaviors, more targeted questions on drug use, condom use, sex work reasons and anal intercourse will be included as well.

**Outcome of interest.** An HIV diagnostic orientation test will be used to assess the presence of antibodies in an oral fluid sample as described by Platteau et al. [44] and Loos et al. [38]. The participants will receive a sampling kit identified with a unique sample code which will be linked to the anonymous behavioral questionnaire, and they will take the oral fluid aliquot following the instructions.

Oral fluid is collected and tested for the presence of HIV antibodies with the DPP HIV1/2 Assay (Chembio Diagnostic Systems, Inc). This assay has been previously evaluated at the Institute of Tropical Medicine by Beelaert et al. [45] and prequalified by WHO [46]. Samples are self-collected by the participants through swabbing of the outer gums with the DPP collecting device. The swab is placed into a sample buffer for shipment to the laboratory. Upon arrival in the lab, the swab is tested with the DPP HIV1/2 assay rapid test. The assay is based on a immunochromatographic test principle, with a test line which reacts positive in case of presence of HIV-antibodies and a control line, which allows detection of human IgG and serves as a built-in control for adequate sample quality and correct performance of the test. Prior to participation, participant registers on the website via an account. In this procedure, the participant provides both an email address and cell phone number. As soon as the result is available, participant receives an email indicating that the test result is accessible. Project collaborators monitor whether participants picked up their test result. In case a participant does not pick up a negative test result, he/she will receive two reminder emails. Participants with a reactive result who do not pick up their test result will be reminded several times via email and phone at a second stage. People with a reactive result, are followed-up for further confirmation testing on a blood sample according to standard-of-care HIV testing algorithm. Follow-up continues until project collaborators receive a confirmation test result. All initial analyses will be carried out by the Institute of Tropical Medicine (Antwerp, Belgium) and the outcome assessor will be blinded to the results of the behavioral questionnaire (risk factors). Confirmation tests will be executed at the Institute of Tropical Medicine, or via existing healthcare services.

**Survey development.** Questionnaires are developed by the research team and tested and adjusted in correspondence with the community advisory group. The research team will make the final decision. Questionnaires will be developed in Dutch and translated into French, Spanish and English, as well as other languages if this appears necessary after the community mapping study. Because of recruitment taking place in a variety of non-research settings, we opted for a relatively short questionnaire to fill in on the spot, as well as a follow-up questionnaire with additional questions which is sent to the participants later by email. The main questionnaire (S1 Appendix) will include demographic questions, questions concerning medical history and questions regarding sexual and drug-using risk behavior and the use of relevant medicines (pre-exposure prophylaxis, PrEP). The follow-up questionnaire (S2 Appendix) will be sent to participants by email no longer than two weeks after filling in the main questionnaire and donating the oral fluid sample. This questionnaire will include additional demographic variables (marital status, living situation, education, financial situation), transgender specific questions (gender affirming hormone use, surgery), sexual health and satisfaction and additional questions regarding risk behavior (drug use, condom use, reasons to engage in transactional sex) and confounding factors (blood or organ transfusion, endoscopic examination).

**Selection and data collection procedures.** After the systematic selection of clusters from the sampling frame, the investigators will randomly select and approach TGNB people from each cluster at predefined time periods. The procedure for selection of participants will be

tailored to the type of setting. During consultation periods at healthcare settings, TGNB people will be randomly selected based on the available lists. At physical gatherings, TGNB people will be recruited from randomly selected tables in bars or at meetings and at parties or public events, potential respondents are randomly approached as they cross a predefined line or enter a predefined area. In digital clusters, we will also randomly select TGNB people based on available list of group/forum members. Only digital clusters that specifically targets TGNB persons will be included in this study ensuring that most of the group members potentially meet the eligibility criteria. Open and closed social media groups or online fora will be targeted. However, for closed groups permission to recruit group members is always asked to the administrators of the group. For privacy reasons, secret groups will not be included whatsoever in this study. The member lists are available in advance for selection reasons only after approval by the group administrators and the study will be briefly introduced by e.g. a wall post. The selected TGNB persons will be contacted for participation by the group administrator or peer data collector with a personal message. If (digital) networks of influencers/key informants are used, the researchers will perform the random selection based on an anonymized list and all communication will exclusively be done by the key informant. In clusters with low number of attendees (ten or less), everyone will be approached for participation.

In healthcare settings, the data will be collected by the main research team while data collection during community events and meetings will also involve peer data collectors, who will be trained and compensated for their time. One of the (peer) data collectors will approach potential participants and introduce the study, its objectives, methodology, informed consent and confidentiality. Interested participants will be invited to a separate room or area (if possible) to register in the online system, complete a structured questionnaire on a tablet and donate an oral fluid sample via a self-collection swab. Informed consent is obtained at the beginning of the questionnaire. Registering on the online platform using a personal email address is necessary to be able to check the HIV test results afterwards. If a test appears reactive, participants are linked to the HIV reference centre or other physician for a confirmation test. Community researchers will also collect data on the characteristics and the reason for refusal of those who refused participation in the study, as well as the overall number of TGNB people present during the onsite data collection.

In digital clusters, after receiving all information concerning participation in the study in a personal message, participants are guided to the online platform to fill in the questionnaire, and are asked to provide an home address to have the oral fluid test kit sent to them which they will have to send back in a prepaid envelope. If selected TGNB people are reluctant to participate, the key informant or peer data collector will make two attempts to find out the reasons for refusal. Similarly, when selected TGNB people do not respond to the invitation, the key informant or peer data collector will send one reminder notification within one week.

Although the above recruitment strategies are directed at gaining a representative study population of TGNB people, it might be difficult to include specific groups within the TGNB community [47]. For clusters in which specific hard-to-reach subpopulations gather (e.g. anarchistic TGNB activists, TGNB refugees involved in transactional sex, TGNB people of color), one can anticipate that it could be difficult to gain access as a researcher. In this case, there is the possibility to include the network of key community members (e.g. personal social media groups) who will likely already have been identified during the formative part of the study. They can act as peer recruiters, randomly contact TGNB people within their hard-to-reach network (herein defined as cluster), informing them about the study and guiding them to the online platform for data collection. In this scenario, the peer recruiters will also collect data on the characteristics and the reason for refusal of those who are reluctant to participate in the study.

**Confidentiality and ethics.**   We will ensure confidentiality by using identification codes to link oral fluid tests and questionnaires. Contact information is needed for sending participants the oral fluid test in case of online recruitment, for the participant to check their testing results, and to ensure linkage to care for participants whose test was reactive. This information is linked to the identification codes in a confidential linking document which can only be consulted by the researcher responsible for data input and the person who is responsible for follow-up and linkage to care. Questionnaires will be pseudonymized upon entry in the database, except for a unique questionnaire and outcome number. The coded personal data will be used for data analysis and the results will be reported anonymously. This study was approved by the Ghent University Hospital Ethics Committee on September 17, 2020 (BC-08157; S1 and S2 Files) and July 28, 2021 (BC-08527; S3 and S4 Files). Written informed consent will be obtained at the beginning of the questionnaire. Potential protocol modifications will be approved by the ethics committee and communicated in the trials register and peer-reviewed publications (cfr. infra). Transparent study procedures that address safety and confidentiality are essential [15, 22, 24]. Hence, we will invest in strong, transparent and clear communication about the study (topic) towards the transgender community throughout all study phases, which will likely increase willingness to participate. This will involve personal communication with community members, organisers and administrators as well as broad social media communication. The data collection was started in December 2021 and anticipated date of completion is December 2022.

**Statistical analysis plan.**   The proportion of HIV infection and 95% confidence interval will be estimated adjusting for clustering and unequal probability of selection. TGNB people who visited sites more frequently had a higher probability of selection in the study. (Individual) weighting factors will be calculated based on the attendance information provided by the participant to adjust for this unequal selection probability. Moreover, the probability of accepting participation and the sampling fraction (number of participants out of total number of eligible attendees at the cluster) will also be taken into account. One can expect that the real number of attendees at outreach events may deviate from the expected number used for the systematic sampling procedure. The actual number of eligible TGNB persons will always be reported in the data collection form and, if needed, weighting factors will be calculated to correct for the potential deviation in sampling probability.

Socio-demographic and geographic characteristics, sexual and drug-related risk factors of the sample will first be summarized by descriptive statistics. Logistic regression models taking into account clustering and using the sampling weights (being inversely proportional to the selection probability) will be fitted to calculate the odds ratio (OR) and 95% CI for each possible risk factor. Nine variables will be queried to map sexual risk behavior among TGNB people and three variables deal with the use of needles or risky blood contacts in the past. Depending on the number of TGNB people with HIV infection, construction of a multivariable model will be considered to estimate adjusted OR (aOR). After constructing a logistic regression model including all confounding factors (e.g. age or country of birth) that had a P<0.05 in univariate analysis, the significant risk factor(s) will then be separately added as explanatory factors to estimate aOR when deemed appropriate. Statistical analyses will be undertaken using the package 'survey' in RStudio to fit the above logistic regression models. Marginal models in the 'survey' package focus on the population mean response, averaged over all clusters, taking into account clustering and the sampling weights. Sensitivity analyses might be performed to assess the impact of e.g. potential protocol violations on the robustness of the results.

**Assessment of potential study bias.**   *Appropriate eligibility criteria*. In contrast to using a convenience sample, TLS allows the recruitment of a more representative study population by approximating random cluster sampling. Against this background, we will ensure that the

sample frame will be completed by interviewing a wide range of key informants in terms of gender identity, ethnicity, age and sexual orientation and until saturation occurs. To ensure diversity in healthcare settings, not only large multidisciplinary gender teams will be included, but also independent professionals working in transgender healthcare who are listed on the Transgender Healthcare Map. If necessary, the sampling frame will be updated during the course of the project to avoid the pitfall of missing some settings. Nevertheless, there will be selection bias towards those who need gender affirming medical care and attend the identified events and locations while TGNB people who never or rarely attend have almost zero probability of being sampled. To (partly) address this limitation, online digital spaces will also be identified and included in the sampling frame. Moreover, we propose that the TLS data will be analysed as a two-stage sample survey using a simple weighting procedure based on the inverse of the approximate probability that a person was sampled. Non-response analysis will also be performed to detect differences regarding gender and age between those who participated and those who declined, and unequal probability of accepting participation will be taken into account when calculating weighting factors.

Selection bias can also be introduced if community researchers do not adhere carefully to design and implementation procedures. There will only be a representative sample if there is no deviation from sampling strategies and interviewing respondents for each place and time. This is why all peer data collectors who are involved in field testing will be trained by the main research team to recruit respondents in the same way. A standard operating procedure on recruitment of participants and data collection will be developed paying special attention to potential bias and inclusive and cultural sensitivity.

To roughly verify whether a representative sample was acquired based on the used sampling strategy, the demographic characteristics (age and gender) of the study population will be compared with the current figures on demographic composition of the TGNB population in Flanders and Belgium. However, this comparison should be interpreted with caution since the group of TGNB people who are visible in the current statistics, are probably only the tip of the iceberg

*Appropriate methods for exposure and outcome variables*. Data on exposure of interest will retrospectively be collected through self-administered questionnaires. The quality of the data is therefore determined to a large extent by the person's ability to accurately recall past exposures to risk factors. Recall bias may result in either an underestimate or overestimate of the association between exposure and outcome. HIV positive participants are often more likely to recall exposure to risk factors than the healthy controls. Moreover, self-reporting data can also be affected by social desirability or approval since the questionnaire includes private and sensitive topics. As a consequence, the results on sexual or drug-using risk behavior could be underreported. Therefore, it is crucial to guarantee confidentiality at the time of data collection [48]. Although participants who are aware of the exact research question are much more likely to cloud their opinions and memories about the risk factors, we will not be able to blind the participants to the study hypothesis for ethical reasons and for guaranteeing a full transparent process to the community. It will also not be possible to blind the peer data collectors to the study hypothesis but they will be trained in data collection and need to adhere to a standardized operating procedure. Furthermore, the peer data collectors will be blinded to the outcome variable (HIV infection). Outcome variables will be collected from saliva analyses with objective and clear diagnostic criteria and the outcome assessors will be blinded to the results of the questionnaire.

*Controlled for confounding*. If possible, multivariable logistic regression analysis will be used to control for all confounding factors at the same time (cfr. supra). Factors that are expected to be related with the risk factors of interest and/or HIV infection are age, gender identity, socio-

economic status, medical history, ethnicity and (parent's) country of birth (whether or not born in an HIV endemic country).

*Complete and adequate follow-up and handling missing data.* Exposure and outcome variables will be collected at the same time and no prospective follow-up of participants is foreseen in this study protocol. Of note, one can assume that the response rate for the follow-up questionnaire would be relatively low but all critical study variables are surveyed at the time of data collection. However, participants can be excluded post-test (i) when they do not belong to the target group (e.g. MSM population, residing outside Flanders or Brussels or < 18 years) based on the self-reported questionnaire or (ii) when data were wrongly entered or linked in the database. It is of note that TGNB people who refused to provide an oral fluid sample after completing the questionnaire will be included to detect any systematic difference in their characteristics and risk behavior with those who completed all steps of the study. Since TGNB people with a reactive oral fluid sample will be invited for further confirmation testing on a blood sample, there is also a risk of missing confirmed outcome data. We assure that we will do all within our capabilities to minimize the loss-to-follow-up via intensive follow-up by email and phone. At this stage, we cannot predict the loss of participants for the confirmation testing and, in first instance, both the reactive oral fluid samples as well as the confirmed HIV status will be assessed as outcome variables. For assessing the impact of the missing data on the study results, multiple imputation will be considered as method for adjusting for missing data [49].

**Dissemination.**   This study was registered at Clinicaltrials.gov (NCT04930614). The results of this study will first of all be submitted to a peer-reviewed journals within 12 months of the final data collection date. However, researchers should ensure their research is framed in a way that positively impacts transgender health and minimizes possible harmful utilisation of the results [25]. The results of this study should ultimately benefit transgender health without increasing stigma and discrimination and with respect for the TGNB community. It is advised that, instead of only focussing on academic dissemination of findings, research should be disseminated back to the community and should be accessible to the TGNB community and non-academic healthcare professionals [15, 25]. Therefore, after publication of the results in peer-reviewed journals, the results will be disseminated to a broad public as well as the TGNB community itself by using the general communication channels of the Institute for the Equality of Women and Men (IGVM) and the Transgender Information Point—the Flemish government-funded expertise centre on transgender issues—as well as the social media channels that were set up for the study. The findings of the study will be available in the form of a report as well as a range of infographics and web pages. The researchers will also actively engage with mainstream and transgender specific media channels to establish the intended interpretation of the study results.

## Discussion

Although HIV prevalence has been estimated relatively high in transgender people (roughly 12–28%), current literature has mainly focused on high risk groups, such as female transgender sex workers, and thereby largely neglected intra-group variation within the TGNB community. Up until now, large well-designed studies investigating HIV prevalence and associated risk factors in this heterogeneous population based on HIV diagnostic orientation testing are completely lacking. The protocol of this study was conceptualized to provide the first estimate of HIV prevalence in a representative sample of TGNB people in Flanders and Brussels by combining CBPR and two-stage TLS strategies. The application of this unique combination will allow us to generate a large and broad sampling frame and to systematically collect data from TGNB people attending the clusters in the sampling frame. By associating HIV testing

results with the data extracted from behavioral questionnaires, this cross-sectional study not only aims to estimate the HIV prevalence rate in TGNB people residing in a Western European country but also to identify high risk groups within this TGNB community. Such data is needed to develop effective targeted sexual health interventions for TGNB people as well as avoid to stigmatizing and discriminating policies.

The approach of the study presents a number of challenges. First, to obtain an accurate representation of the TGNB population in Flanders and Brussels it is crucial to develop an exhaustive sampling frame of (online) community and healthcare settings covering all aspects of social life in order to minimize the probability of systematically missing target groups. Moreover, another potential limitation of this study is that the assessment of risk factors rely on participants recall to exposed risks in the past 12 months. Lastly, implementing the CBPR approach in this study will be challenging in terms of assuring skills of lay researchers and guaranteeing motivation to adhering to the protocol and data quality [50, 51]. However, the CBPR approach will help us to reach hidden TGNB subgroups and contribute to the acceptability of the study and its methods by the community. To provide a solution for the potential pitfalls, standard operating procedures on participant recruitment, data collection and data quality will be developed to train and coach the lay researchers.

The strength of this study is that it will directly expand our knowledge base and understanding of HIV prevalence rates among the hard-to-reach and vulnerable TGNB population. By systematically identifying all relevant clusters where TGNB people can be reached, and by randomly sampling TGNB people at the sampling clusters, we will be able to generate a representative sample of TGNB people and, hence, to generalize the study findings to the well-defined TGNB population in Flanders and Brussels or even Western Europe. For hard-to-reach populations such as TGNB people, TLS is the most effective method for obtaining probability samples of populations who can be located at well-defined venues.

## Supporting information

**S1 Checklist. SPIRIT checklist.**
(DOC)

**S1 Appendix. Questionnaire survey.**
(PDF)

**S2 Appendix. Follow-up questionnaire survey.**
(PDF)

**S1 File. Application form (BC-08157) that was approved by the Ghent University Hospital Ethics Committee.**
(DOCX)

**S2 File. Translated application form (BC-08157) that was approved by the Ghent University Hospital Ethics Committee.**
(DOCX)

**S3 File. Application form (BC-08527) that was approved by the Ghent University Hospital Ethics Committee.**
(DOCX)

**S4 File. Translated application form (BC-08527) that was approved by the Ghent University Hospital Ethics Committee.**
(DOCX)

## Author Contributions

**Conceptualization:** Niels De Brier, Judith Van Schuylenbergh, Hans Van Remoortel, Emmy De Buck, Guy T'Sjoen, Veerle Compernolle, Tom Platteau, Joz Motmans.

**Funding acquisition:** Guy T'Sjoen, Joz Motmans.

**Methodology:** Niels De Brier, Judith Van Schuylenbergh, Hans Van Remoortel, Dorien Van den Bossche, Steffen Fieuws, Geert Molenberghs, Emmy De Buck.

**Project administration:** Judith Van Schuylenbergh, Hans Van Remoortel.

**Supervision:** Hans Van Remoortel, Emmy De Buck, Guy T'Sjoen, Veerle Compernolle, Tom Platteau, Joz Motmans.

**Validation:** Niels De Brier, Judith Van Schuylenbergh, Hans Van Remoortel.

**Writing – original draft:** Niels De Brier, Judith Van Schuylenbergh.

**Writing – review & editing:** Hans Van Remoortel, Dorien Van den Bossche, Steffen Fieuws, Geert Molenberghs, Emmy De Buck, Guy T'Sjoen, Veerle Compernolle, Tom Platteau, Joz Motmans.

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
