## [Decision Letter · Decision Letter 0]

11 Jan 2022

PONE-D-21-25105

Prevalence and associated risk factors of HIV infections in a representative transgender and non-binary population in Flanders and Brussels (Belgium): Protocol for a community-based, cross-sectional study using time-location sampling

PLOS ONE

Dear Dr. De Brier,

Thank you for submitting your manuscript to PLOS ONE. After careful consideration, we feel that it has merit but does not fully meet PLOS ONE’s publication criteria as it currently stands. Therefore, we invite you to submit a revised version of the manuscript that addresses the points raised during the review process.

We look forward to receiving your revised manuscript.

Kind regards,

Hong-Van Tieu

Academic Editor

PLOS ONE

https://journals.plos.org/plosone/s/file?id=ba62/PLOSOne_formatting_sample_title_authors_affiliations.pdf”

Additional Editor Comments:

Thank you for submitting this manuscript “Prevalence and associated risk factors of HIV infections in a representative transgender and non-binary population in Flanders and Brussels (Belgium): Protocol for a community-based, cross-sectionals study using time-location sampling” which seeks to recruit a representative sample of transgender and non-binary adults in Belgium and determine the HIV prevalence in the community. The reviewers found that this topic represents important work and roadmap for creating representative samples of TGNB people in other regions. Reviewer comments are included, which will require revision for consideration for acceptance for publication.

Reviewers' comments:

Reviewer's Responses to Questions

**Comments to the Author**

1. Does the manuscript provide a valid rationale for the proposed study, with clearly identified and justified research questions?

Reviewer #1: Yes

Reviewer #2: Yes

2. Is the protocol technically sound and planned in a manner that will lead to a meaningful outcome and allow testing the stated hypotheses?

Reviewer #1: Yes

Reviewer #2: Yes

3. Is the methodology feasible and described in sufficient detail to allow the work to be replicable?

Reviewer #1: Yes

Reviewer #2: Yes

4. Have the authors described where all data underlying the findings will be made available when the study is complete?

Reviewer #1: Yes

Reviewer #2: Yes

5. Is the manuscript presented in an intelligible fashion and written in standard English?

Reviewer #1: Yes

Reviewer #2: Yes

6. Review Comments to the Author

You may also provide optional suggestions and comments to authors that they might find helpful in planning their study.

Reviewer #1: Very competent and detailed report of planned study methods for a survey to estimate HIV prevalence in transgender and non-binary people in Flanders and assess associated risk factors.

There are two points where more information is needed, in my opinion.

(A) Calculation of the sizes of clusters and thus the sampling fractions. Line 554 ‘Sampling fraction (number of participants out of total number of eligible attendees at the cluster)’ – how will the total eligible numbers be determined or estimated for these diverse types of clusters (‘medical, mental health and social services, pride events, bars, discussion groups, social media groups, online fora,’)? Especially as ‘eligible’ means TGNB, is this realistic? Cluster sizes are required for the systematic sampling of clusters and for weighting during estimation.

(B) Line 491: Selection of people in digital clusters is ‘based on available list of group/forum members’ – do such lists exist for all these various types of cluster? Will it be possible to create the list of TGNB people in advance, or will this eligibility criterion be determined by questioning the individual people?

In view of the potential biases associated with the TLS sampling method, which are appropriately discussed by the authors, I wondered whether a simple random sample from the Flanders population might be feasible and maybe more reliable. According to the figure given in lines 267-272, around 40 000 questionnaires would need to be sent out in order to find 1000 TGNB people. This is of course only possible if complete lists of inhabitants with contact details are accessible. Even if this alternative method cannot be fully implemented, a smaller such sample might serve as a check on the completeness and representativeness of the main TLS sampling method. Could the authors comment on this?

Minor points:

1. Line 320-321: The proportions with HIV are p0, p1 (lower-case), while the proportions with and without the risk factor are P0, P1 (upper-case) – so in these lines it should be lower-case p0, p1?

2. Line 322 ‘Since the risk factor will be assessed at the level of the individual TGNB people, the use of the design effect for clustering is a conservative approach here.’ – is that so?

3. Line 566 Logistic models are fitted ‘incorporating clustering’ – does this mean that cluster is fitted as an explanatory factor and therefore estimates of HIV prevalence per cluster will be calculated? Or does it mean that a kind of ‘mixed model’ is used to allow for random cluster effects?

4. Line 675 ‘The TLS methodology will not allow for inference from a geographically or demographically defined sample’ – meaning unclear to me.

5. A few minor language corrections need to be made, e.g. lines 467 'into French', 491 'time slots', 560 'nine variables will be questioned': do you mean queried or analysed?, 595 'determined to a large extent by', 602 'as well as possible. In line 563 do you mean 'confounding factors' or 'explanatory factors'?

Reviewer #2: The study “Prevalence and associated risk factors of HIV infections in a representative transgender and non-binary population in Flanders and Brussels (Belgium): Protocol for a community-based, cross-sectionals study using time-location sampling” seeks to recruit a representative sample of transgender and non-binary (TGNB) adults in Belgium and determine the HIV prevalence in this community. Given the nascent nature of public health-related research with this population, and its to-date focus on high-risk members of the community (including sex workers) or clinic-based samples. As such, this is important work and can help create a roadmap for creating representative samples of TGNB people in other places. I do have some concerns, however.

A general note: Throughout the paper there are slight translation and idiomatic issues that I think could use the benefit of an editor. For example, in line 38, the phrase “transgender women being people” reads more easily as “transgender women are people” with a similar comment for transgender men on the next line (line 39). Another example can be found in line 85, “It becomes clear that as long as gender identity is not taking into account” should read “It becomes clear that as long as gender identity is not taken into account”. Finally, in line 602, “information bias should be as good as possible” should be “information bias as well as possible”.

Major Concerns

1. In a number of places in the introduction, you use the phrase “HIV prevalence has been estimated high for transgender people” or similar phrasing. I think you would make a stronger argument if you introduced the prevalence ranges you cite earlier in the introduction. Further, it is not clear (without presenting the data) if you are making absolute statements or comparing HIV prevalence to other groups.

2. In lines 55-58, the language that you use to describe possible mechanisms for the effects of stigma and discrimination feels stigmatizing. Facing stigma and discrimination are associated with HIV risk behavior but focusing solely on potential individual-level mechanisms (low self-esteem, mental health, a need for affirmation from others) instead of more structural factors (or ignoring the structural factors altogether) reads as a stigmatizing statement. Perhaps TGNB who face discrimination cannot negotiate safe sex because of (or in part) marginalization. They could lack the power to negotiate safer sex. The structural issues could be less of a factor in Belgium than in other countries, but this part of the introduction reads as a more global comment.

3. One of the arguments made is that most research is conducted among transgender women. Some quantification of this (you use the phrase “a lot” in line 65) would be helpful. What proportion of studies focus on transgender women? Further, in the next sentence, the point is made that most research among TGNB is conducted in large sexual health clinics and community centers, which leads to samples that are disproportionately at high risk for HIV. This seems to be missing citations.

4. In the introduction the authors criticize other work for taking place mainly in large, urban, population centers. Yet, in the TLS scheme, the authors select Brussels and Flanders rather than a broader survey of Belgium. Some (even brief) discussion of how much of the population is included in this region or some of the rationale for location choices would be helpful. Especially as the objective is to create a representative sample of TGNB residing in a Western European country. The addition of digital LGBT+/TGNB spaces may penetrate into other regions, but some discussion would be clarifying.

5. The authors refer to their study as a CBPR study, but it is unclear exactly what that means. While it is undoubtedly a community-based study, CBPR requires more and it is not clear what leadership or major roles members of the TGNB community or supporting organizations actually play. For example, are members of the TGNB study co-investigators or parts of a Community Advisory Board (CAB)? Did they play a role in the design of the study or survey? In what ways are members of the community driving the research or research question? How are members of the community partners in this study as opposed to participants only?

6. In the methods section, it is unclear why a cluster is expected to be on average 18 TGNB. This seems large for a cluster, even for estimation purposes. I think that a hypothetical situation (a single event or cluster would be defined as xxx location for yyy hours, yielding approximately zzz contacts) would be helpful.

7. In lines 353-354, the authors state that the number of TGNB…will be fixed at 50% of the total…events. Does this mean that 50% of TGNB particopants will be recruited from healthcare settings and the remaining 50% from outreach events? It currently reads as though TGNB will be 50% of those sampled.

8. In the description of variables section, the terms transvestite and cross-dresser are offered as gender identity options. This may be a country-specific issue (different terms used across countries), but these terms seem both anachronistic and not to measure gender identity.

9. In the selection and data collection procedures section (lines 491), it is unclear how privacy issues around data collection will be handled in the digital clusters.

10. In the section that discusses appropriate methods for exposure and outcome variables, the authors state that participants will be blinded to the study hypothesis to avoid social desirability responses. While I understand the desire to minimize information bias, this seems to be both problematic from an ethical standpoint (informed consent) but also stands in opposition to the authors statements throughout the paper of creating a transparent process. It is also unclear how the authors propose to do this since participants are also giving a saliva sample.

Minor Concerns

1. In line 440, it is unclear what an HIV orientation test is, this reads like a typo.

2. In lines 490-491, “pre-determined time sloths” should read “pre-determined time slots”

7. PLOS authors have the option to publish the peer review history of their article (what does this mean?). If published, this will include your full peer review and any attached files.

Reviewer #1: **Yes: **Jeremy Franklin

Reviewer #2: No

---

## [Author Response · Author response to Decision Letter 0]

10 Feb 2022

Reply to the remarks of the reviewers of manuscript ID PONE-D-21-25105

Dear Prof. Dr. Hong-Van Tieu

Dear Editor

Thank you for the work done on manuscript ID PONE-D-21-25105 entitled “Prevalence and associated risk factors of HIV infections in a representative transgender and non-binary population in Flanders and Brussels (Belgium): Protocol for a community-based, cross-sectional study using time-location sampling” by Niels De Brier, Judith Van Schuylenbergh, Hans Van Remoortel, Dorien Van den Bossche, Steffen Fieuws, Geert Molenberghs, Emmy De Buck, Guy T'Sjoen, Veerle Compernolle, Tom Platteau and Joz Motmans. 

Reviewers commented positively on our submission and had some comments and suggestions for improving the manuscript. We now have revised the manuscript and believe that we have upgraded its quality. Below, we detail how we have addressed all of the individual points raised. We hope that the revisions made and responses given are clear such that the manuscript will now be acceptable for publication. 

Yours sincerely,

Niels De Brier, corresponding author on behalf of all co-authors. 

Journal requirements

https://journals.plos.org/plosone/s/file?id=ba62/PLOSOne_formatting_sample_title_authors_affiliations.pdf”

We doublechecked the journal guidelines and ensured that the manuscript meets the PLOS ONE's style requirements. All first lines of new paragraphs in the manuscript are now indented. 

The ethics statement is only elucidated in the Methods section in lines 580-598.

Additional Editor Comments

Thank you for submitting this manuscript “Prevalence and associated risk factors of HIV infections in a representative transgender and non-binary population in Flanders and Brussels (Belgium): Protocol for a community-based, cross-sectionals study using time-location sampling” which seeks to recruit a representative sample of transgender and non-binary adults in Belgium and determine the HIV prevalence in the community. The reviewers found that this topic represents important work and roadmap for creating representative samples of TGNB people in other regions. Reviewer comments are included, which will require revision for consideration for acceptance for publication.

We thank the editor for the work done on the manuscript, for her positive comments on our research protocol and for providing clear guidance on how to upgrade the quality of the manuscript. 

Reviewer #1

Very competent and detailed report of planned study methods for a survey to estimate HIV prevalence in transgender and non-binary people in Flanders and assess associated risk factors.

We thank the reviewer for his/her positive comment.

There are two points where more information is needed, in my opinion.

(A) Calculation of the sizes of clusters and thus the sampling fractions. Line 554 ‘Sampling fraction (number of participants out of total number of eligible attendees at the cluster)’ – how will the total eligible numbers be determined or estimated for these diverse types of clusters (‘medical, mental health and social services, pride events, bars, discussion groups, social media groups, online fora,’)? Especially as ‘eligible’ means TGNB, is this realistic? Cluster sizes are required for the systematic sampling of clusters and for weighting during estimation.

To address this comment, we now elaborated on how we will estimate the cluster sizes in more detail in lines 412-418 as follows: “For determining the number of attendees in health care facilities, the exact number of patients attending consultation periods at specific time slots will be provided by the counsellors or the hospital data managers for the upcoming three months. Regarding the outreach events (e.g. pride events, parties or gatherings in bars, discussion groups, …), the organizers will be contacted for collecting data on the expected number of TGNB persons. In social media groups or online fora, the researchers will have access to the full list of members after approval by the group administrators.”

Moreover, in lines 605-609, we now detailed how we mitigate potential deviations from the expected number of attendees: “One can expect that the real number of attendees at outreach events may deviate from the expected number used for the systematic sampling procedure. The actual number of eligible TGNB persons will always be reported in the data collection form and, if needed, weighting factors will be calculated to correct for the potential deviation in sampling probability.” 

(B) Line 491: Selection of people in digital clusters is ‘based on available list of group/forum members’ – do such lists exist for all these various types of cluster? Will it be possible to create the list of TGNB people in advance, or will this eligibility criterion be determined by questioning the individual people?

We thank the reviewer for his/her comment and now specified the selection of TGNB people in digital clusters in lines 539-548 as follows (see also comment 9 of reviewer 2 below): “Only digital clusters that specifically targets TGNB persons will be included in this study ensuring that most of the group members potentially meet the eligibility criteria. Open and closed social media groups or online fora will be targeted, however for closed groups permission to recruit group members is always asked to the administrators of the group. For privacy reasons, secret groups will not be included whatsoever in this study. The member lists are available in advance for selection reasons only after approval by the group administrators and the study will be briefly introduced by e.g. a wall post. The selected TGNB persons will be contacted for participation by the group administrator or peer data collector with a personal message. If (digital) networks of influencers/key informants are used, the researchers will perform the random selection based on an anonymized list and all communication will exclusively be done by the key informant.”

In view of the potential biases associated with the TLS sampling method, which are appropriately discussed by the authors, I wondered whether a simple random sample from the Flanders population might be feasible and maybe more reliable. According to the figure given in lines 267-272, around 40 000 questionnaires would need to be sent out in order to find 1000 TGNB people. This is of course only possible if complete lists of inhabitants with contact details are accessible. Even if this alternative method cannot be fully implemented, a smaller such sample might serve as a check on the completeness and representativeness of the main TLS sampling method. Could the authors comment on this?

We agree with the reviewer that simple random sampling could be an alternative approach but an accurate statistical measure of the TGNB population can indeed only be obtained when a full list of the entire population to be studied is available. However, there is no population data regarding gender identity in Belgium. We could only select TGNB persons based on official gender change information when using national registry data, but gaining access to the full list of TGNB persons in Brussels and Flanders presents major challenges due to legal constraints and privacy policies. Furthermore, this would imply a direct and serious selection bias since only 2734 persons changed their legal gender marker in Belgium between 1993 and 2020. Until 2018, TGNB persons needed to have undergone sterilization to change their gender marker. Since 2018, this is no longer the case, but legal gender markers are still binary (i.e. no option for non-binary persons). It is important to note that not all TGNB persons feel dissatisfied with their own body, or have a desire for medical or/or legal gender reassignment. Using a population sample based on a change in gender marker would thus not lead to a representative sample of TGNB persons. Finally, cluster sampling is time- and cost-efficient, especially for samples that are widely geographically spread and would be practically difficult to properly sample otherwise. 

As a potential way to roughly check the representativeness of our sampling method, the demographic characteristics of our study population will be compared with the current view on demographic composition of the TGNB population in Belgium (and Europe). In the unlikely case that large and profound deviations will be detected, one can consider demographic weighting but it would be difficult to reliably match the demographics due to uncertainties in the current statistics of TGNB people in Flanders and Brussels. In response, the sentences in lines 648-652 now read as follows: “To roughly verify whether a representative sample was acquired based on the used sampling strategy, the demographic characteristics (age and gender) of the study population will be compared with the current figures on demographic composition of the TGNB population in Flanders and Belgium. However, this comparison should be interpreted with caution since the group of TGNB people who are visible in the current statistics, are probably only the tip of the iceberg.”

Minor points:

1. Line 320-321: The proportions with HIV are p0, p1 (lower-case), while the proportions with and without the risk factor are P0, P1 (upper-case) – so in these lines it should be lower-case p0, p1?

We thank the reviewer for noting this difference and changed the spelling of p0 and p1 to lower-case.

2. Line 322 ‘Since the risk factor will be assessed at the level of the individual TGNB people, the use of the design effect for clustering is a conservative approach here.’ – is that so?

This design effect here depends on whether the risk factor will be measured at cluster level or at subject level. At the cluster level, the design effect is the same as with the prevalence estimate but since we will map the risk factors at the subject level, the design effect will be smaller since the risk factor of interest varies within a cluster. In response, the sentences now read as follows in lines 362-365: “Since the risk factor will be assessed at the level of the individual TGNB person, the use of the design effect for clustering is a conservative approach here. Indeed, the design effect will be smaller than with the prevalence estimate since the risk factor of interest varies within a cluster.”

3. Line 566 Logistic models are fitted ‘incorporating clustering’ – does this mean that cluster is fitted as an explanatory factor and therefore estimates of HIV prevalence per cluster will be calculated? Or does it mean that a kind of ‘mixed model’ is used to allow for random cluster effects?

We will not use a multilevel/hierarchical model, where the clusters are the random effect but we will rather apply a marginal model similar to the PROC SURVEYLOGISTIC in SAS which take into account both the sampling weight and clustering. The sampling weight will affect the calculation of the point estimate, and the clustering will affect the calculation of the standard errors. The clusters are not added as explanatory factors. Nevertheless, certain cluster characteristics such as type of event of organizer can be added as covariates when deemed appropriate. To address this comment, we further elaborated on the logistic models as follows in lines 619-622: “Statistical analyses will be undertaken using the package ‘survey’ in RStudio to fit the above logistic regression models. Marginal models in the ‘survey’ package focus on the population mean response, averaged over all clusters, taking into account clustering and the sampling weights.”

4. Line 675 ‘The TLS methodology will not allow for inference from a geographically or demographically defined sample’ – meaning unclear to me.

We agree with the reviewer and rephrased the sentence as follows in lines 737-739: “For hard-to-reach populations such as TGNB people, TLS is the most effective method for obtaining probability samples of populations who can be located at well-defined venues.”

5. A few minor language corrections need to be made, e.g. lines 467 'into French', 491 'time slots', 560 'nine variables will be questioned': do you mean queried or analysed?, 595 'determined to a large extent by', 602 'as well as possible. In line 563 do you mean 'confounding factors' or 'explanatory factors'?

We thank the reviewer for his/her comment and the corrections are readily taken care off as follows:

- In lines 513-515: “Questionnaires will be developed in Dutch and translated into French, Spanish and English, as well as other languages if this appears necessary after the community mapping study.”

- In lines 538-539: Time slots was removed from the following sentence: “In digital clusters, we will also randomly select TGNB people based on available list of group/forum members.” 

- In lines 613-615: “Nine variables will be queried to map sexual risk behavior among TGNB people and three variables deal with the use of needles or risky blood contacts in the past.”

- In lines 654-655: “The quality of the data is therefore determined to a large extent by the person’s ability to accurately recall past exposures to risk factors.”

- In lines 661-664: “as well as possible” was deleted from this sentence based on comment 10 of reviewer 2

- In lines 616-619: “After constructing a logistic regression model including all confounding factors (e.g. age or country of birth) that had a P<0.05 in univariate analysis, the significant risk factor(s) will then be separately added as explanatory factors to estimate adjusted odds ratios (aOR) when deemed appropriate.”

Reviewer #2 

The study “Prevalence and associated risk factors of HIV infections in a representative transgender and non-binary population in Flanders and Brussels (Belgium): Protocol for a community-based, cross-sectionals study using time-location sampling” seeks to recruit a representative sample of transgender and non-binary (TGNB) adults in Belgium and determine the HIV prevalence in this community. Given the nascent nature of public health-related research with this population, and its to-date focus on high-risk members of the community (including sex workers) or clinic-based samples. As such, this is important work and can help create a roadmap for creating representative samples of TGNB people in other places. I do have some concerns, however.

We thank the reviewer for his/her positive comment.

A general note: Throughout the paper there are slight translation and idiomatic issues that I think could use the benefit of an editor. For example, in line 38, the phrase “transgender women being people” reads more easily as “transgender women are people” with a similar comment for transgender men on the next line (line 39). Another example can be found in line 85, “It becomes clear that as long as gender identity is not taking into account” should read “It becomes clear that as long as gender identity is not taken into account”. Finally, in line 602, “information bias should be as good as possible” should be “information bias as well as possible.”

We thank the reviewer for highlighting the grammatical improvements and incorporated the corrections in the text as follows:

- In lines 38-41: “Transgender people are people who do not or to a lesser extent identify with their birth-assigned gender: transgender women are people who were assigned male at birth (AMAB) but identify on the female spectrum; transgender men being people who were assigned female at birth (AFAB) but identify on the male spectrum.”

- In lines 89-90: “It becomes clear that as long as gender identity is not taken into account in general population research, obtaining a generalizable sample of transgender 

people is nearly impossible [16-18].”

- In lines 661-664: “as well as possible” was deleted from this sentence based on comment 10 of reviewer 2.

Major Concerns

1. In a number of places in the introduction, you use the phrase “HIV prevalence has been estimated high for transgender people” or similar phrasing. I think you would make a stronger argument if you introduced the prevalence ranges you cite earlier in the introduction. Further, it is not clear (without presenting the data) if you are making absolute statements or comparing HIV prevalence to other groups.

In response, we added the introduced prevalence rates to the following sentences and highlighted that the rates are relatively high as follows: 

- In lines 99-101: “Although HIV prevalence has been estimated relatively high for transgender women (0-49.6%), HIV prevalence is estimated much lower in transgender men, ranging between 0 and 8.3% [5,6,10], however few studies include or specifically focus on this subpopulation.”

- In lines 707-709: “Although HIV prevalence has been estimated relatively high in transgender people (roughly 12-28%), current literature has mainly focused on high risk groups, such as female transgender sex workers, and thereby largely neglected intra-group variation within the TGNB community.”

For relative comparison reasons, the HIV prevalence rates of the general population has been introduced in lines 32-33. 

2. In lines 55-58, the language that you use to describe possible mechanisms for the effects of stigma and discrimination feels stigmatizing. Facing stigma and discrimination are associated with HIV risk behavior but focusing solely on potential individual-level mechanisms (low self-esteem, mental health, a need for affirmation from others) instead of more structural factors (or ignoring the structural factors altogether) reads as a stigmatizing statement. Perhaps TGNB who face discrimination cannot negotiate safe sex because of (or in part) marginalization. They could lack the power to negotiate safer sex. The structural issues could be less of a factor in Belgium than in other countries, but this part of the introduction reads as a more global comment.

We agree off course that structural factors such as stigmatization and discrimination are important factors in HIV risk behaviour in transgender people. This is why the sentences above the lines cited by the reviewer elaborate on discrimination leading to engagement in sex work and the association of discrimination with sexual risk behaviour, thus structural factors influencing individual-level factors such as low self-esteem and mental health. We hope our point is clear that transgender persons face vulnerabilities on several levels which all have an influence on their health.

3. One of the arguments made is that most research is conducted among transgender women. Some quantification of this (you use the phrase “a lot” in line 65) would be helpful. What proportion of studies focus on transgender women? Further, in the next sentence, the point is made that most research among TGNB is conducted in large sexual health clinics and community centers, which leads to samples that are disproportionately at high risk for HIV. This seems to be missing citations.

We thank the reviewer for this comment. To clarify, we added a quantification from the most recent review on HIV in transgender persons and citations to lines 67-72 as follows: “Oversampling of transgender people at high risk for HIV appears to be common: a lot of studies are focused on transgender women or (female) transgender sex workers [4,10-12]. For example, the recent systematic review of van Gerwen et al. [10] included 25 studies. All of these studies included transgender women but only 9 included data on transgender men, often in small samples. A lot of transgender HIV research has been conducted in sexual health clinics and community centres in large urban areas, resulting in samples of transgender people at high risk for HIV [4,10-12].”

4. In the introduction the authors criticize other work for taking place mainly in large, urban, population centers. Yet, in the TLS scheme, the authors select Brussels and Flanders rather than a broader survey of Belgium. Some (even brief) discussion of how much of the population is included in this region or some of the rationale for location choices would be helpful. Especially as the objective is to create a representative sample of TGNB residing in a Western European country. The addition of digital LGBT+/TGNB spaces may penetrate into other regions, but some discussion would be clarifying.

To address the first comment we now specified the study area in lines 187-198 as follows: “We will conduct a broad survey in two out of three regions of Belgium (Flanders and Brussels). Flanders is the Dutch-speaking northern region of Belgium and counts approximately 6.6 million inhabitants in 2020. It is characterized by high land take, with many urban and rural centres. The Brussels-Capital Region is located in the central part of Belgium comprising 19 municipalities, including the City of Brussels, which is the capital of Belgium. This region has a population of around 1.2 million [32]. In total, Flanders and Brussels cover about 68% of the total Belgian inhabitants. Furthermore, large differences between the regions exist in terms of language, policies, transgender healthcare and services and the organization and structure of the TGNB community. Due to limited resources and difficulties in penetrating into the TGNB community which is to date not well sociologically mapped, TGNB people residing the French-speaking southern part of Belgium, the Walloon region, are excluded in this study. We believe that this focus will benefit the representativeness of our sample.” 

Regarding the concerns on penetrating other regions, the selection criteria now reads as follows: 

- In lines 315-316: “Finally, eligible TGNB people should reside in Flanders or Brussels at the time of data collection.”

- In lines 677-680: “However, participants can be excluded post-test (i) when they do not belong to the target group (e.g. MSM population, residing outside Flanders or Brussels or < 18 years) based on the self-reported questionnaire or (ii) when data were wrongly entered or linked in the database.”

5. The authors refer to their study as a CBPR study, but it is unclear exactly what that means. While it is undoubtedly a community-based study, CBPR requires more and it is not clear what leadership or major roles members of the TGNB community or supporting organizations actually play. For example, are members of the TGNB study co-investigators or parts of a Community Advisory Board (CAB)? Did they play a role in the design of the study or survey? In what ways are members of the community driving the research or research question? How are members of the community partners in this study as opposed to participants only?

We cannot agree more with the reviewer. Involving TGNB people throughout several phases of the study is highly recommended for research involving hard-to-reach populations such as transgender people. We already briefly elaborated on the roles of the community members in lines 192-202 and 493-502 of the originally submitted manuscript but now elucidated more explicitly on how TGNB people will be involved during the process as a separate paragraph as follows in lines 213-243: 

“Community involvement: advisory board and peer data collectors

Throughout the study, CBPR methods will be used to ensure community involvement. CBPR entails participation of the target community throughout all phases of the research process, and is often used in research involving hard-to-reach populations. CBPR models offer mutually beneficial collaborative partnerships between those who fund, sponsor and implement research and the groups, individuals and communities affected by it [25]. Using CBPR methods, TGNB people become co-producers of knowledge [22]. In concrete terms, we will first set up a community advisory board consisting of a diverse range of TGNB people, which is strongly advised for research involving transgender people [25]. At the time of writing, the community advisory board was already composed by the researchers and consisted of 15 TGNB persons (including transgender women, transgender men, queer and non-binary persons). This advisory board will be consulted throughout several phases of the study. In first instance, the acceptability of the study within the TGNB community was already estimated by the community advisory board and we investigated how TGNB people want to be involved in studies investigating HIV in TGNB people. As a result, the community advisory board will also co-create the branding of the study, validate the community map resulting from the in-depth interviews, provide direct input on the content and comprehensibility of the questionnaires of the study, on the recruitment process and on the communication strategy of the published results. The community advisory board will not take part in outcome assessment and data analyses which will be executed by researchers who will not be involved in data collection. Consulting the community advisory board can take place online as well as offline on a regular basis, depending on the preferences of the community advisory board members, ensuring the accessibility of the setting. 

Furthermore, 60% of the main research team consists of TGNB community members, and in addition three peer data collectors will support the researchers during data collection at community events and meetings during the study. Having TGNB peers involved as part of the research team appears to be a strong indicator for participants’ willingness to participate in transgender health studies [15,22,24]. Furthermore, peer data collectors experience fewer hurdles to overcome mistrust between researchers and hard-to-reach communities. However, the participation of TGNB people in the research team can also create a number of challenges, such as potential bias, blind spots and interpersonal factors [25]. Hence peer data collectors require training to address challenges related to peer-to-peer research interactions, as well as to ensure all participants are treated with respect and culturally competent language is used.”

6. In the methods section, it is unclear why a cluster is expected to be on average 18 TGNB. This seems large for a cluster, even for estimation purposes. I think that a hypothetical situation (a single event or cluster would be defined as xxx location for yyy hours, yielding approximately zzz contacts) would be helpful.

It is important to note that the average cluster size in the calculation corresponds to the average number of TGNB persons sampled per cluster and not to the total number of attendees. Number of interviews to be done in each cluster (i.e. cluster size) should be at least ten but the cluster size is variable. In fact, some large clusters will be selected more than once because their expected number of attendees is a multiple of the step size by which the weighted systematic sample is taken. Multiples of 10 TGNB persons will then be selected in these relative large clusters. At this stage of the study, we have little information on the variability in cluster size and, although, one can assume that, in most clusters, only 10 TGNB persons will be recruited, it is a conservative approach to inflate the cluster size when calculating the design effect. To address this comment, we further elaborated on the average cluster size assumptions for the sample size calculations as follows in lines 338-344: “The cluster sizes will be fixed at multiples of ten (cfr. infra) and we expect that the proportion of small clusters will be relatively large. The average number of TGNB people sampled per cluster will hence be close to ten but the impact of the large events on the average cluster size remains to be revealed in this study. Although an average cluster size of 12 to 14 TGNB persons would be realistic based on theoretical simulations, we prefer to be conservative and to inflate the cluster size in the sample size calculations. When we for example assume on average 18 TGNB people per cluster, the design effect would be 2.36 [=(1+(18-1)*0.08] (varying between 1.51 and 3.72).”

By way of example, a large event such as T-day may welcome 200 TGNB attendees during the day based on the subscriptions and this event will be expanded on the sampling list based on the number of attendees and have a very large chance to be selected (multiple times). Hypothetically, the cluster can be selected four times based on the sampling interval and, in this case, 40 TGNB persons should randomly be recruited. Similarly, a discussion group with typically 15 TGNB persons gathering during an evening is a small cluster and have a relatively low chance to be selected. When the cluster will be selected, only 10 TGNB persons of the discussion group will be selected.

7. In lines 353-354, the authors state that the number of TGNB…will be fixed at 50% of the total…events. Does this mean that 50% of TGNB participants will be recruited from healthcare settings and the remaining 50% from outreach events? It currently reads as though TGNB will be 50% of those sampled.

Indeed, we mean that 50% of TGNB participants will be recruited from healthcare settings and 50% from physical and digital outreach events. To avoid confusion, we rephrased this sentence as follows in lines 594-596: “Therefore, at this stage, 50% of TGNB participants in this study will be recruited from healthcare settings and 50% from physical and digital outreach events.”

8. In the description of variables section, the terms transvestite and cross-dresser are offered as gender identity options. This may be a country-specific issue (different terms used across countries), but these terms seem both anachronistic and not to measure gender identity.

The most important criteria for this study is that self-identified TGNB people are included. In Belgium, ‘transgender’ is regarded as an umbrella term which covers a wide range of identities. We know from previous Belgian studies conducted by our research team that cross-dresser and transvestite are indeed terms (a minority of) transgender people use to describe their gender identity in our region. Transvestite or crossdressing people who identify as transgender are hence also included. To further clarify and motivate the inclusion criteria, we now added the following sentence to lines 309-310: “Previous research demonstrated that cross-dresser and transvestite are indeed terms (a minority of) transgender people use to describe their gender identity in Belgium [37].”

9. In the selection and data collection procedures section (lines 491), it is unclear how privacy issues around data collection will be handled in the digital clusters.

To address this comment, we now included the following information in lines 539-548: “Only digital clusters that specifically targets TGNB persons will be included in this study ensuring that most of the group members potentially meet the eligibility criteria. Open and closed social media groups or online fora will be targeted. However, for closed groups permission to recruit group members is always asked to the administrators of the group. For privacy reasons, secret groups will not be included whatsoever in this study. The member lists are available in advance for selection reasons only after approval by the group administrators and the study will be briefly introduced by e.g. a wall post. The selected TGNB persons will be contacted for participation by the group administrator or peer data collector with a personal message. If (digital) networks of influencers/key informants are used, the researchers will perform the random selection based on an anonymized list and all communication will exclusively be done by the key informant.”

10. In the section that discusses appropriate methods for exposure and outcome variables, the authors state that participants will be blinded to the study hypothesis to avoid social desirability responses. While I understand the desire to minimize information bias, this seems to be both problematic from an ethical standpoint (informed consent) but also stands in opposition to the authors statements throughout the paper of creating a transparent process. It is also unclear how the authors propose to do this since participants are also giving a saliva sample.

We fully agree with the reviewer. Although one can advise to blind the participants as well as possible for the study objectives to limit information bias, it will be very difficult to achieve this in the field. Since we definitely do not want the compromise on the transparency of our research project, the sentence in lines 661-664 now reads as follows: “Although participants who are aware of the exact research question are much more likely to cloud their opinions and memories about the risk factors, we will not be able to blind the participants to the study hypothesis for ethical reasons and for guaranteeing a full transparent process to the community.”

Minor Concerns

1. In line 440, it is unclear what an HIV orientation test is, this reads like a typo.

In case of a non-reactive test result, the result is decisive negative. There is no need to retest, or confirm the result. In case of a reactive result, it is important to execute state of the art confirmatory HIV testing on a blood sample using validated laboratory protocols. A reactive result is an indication for HIV positive status, but does not have the value of a formal HIV diagnosis. There is consensus to refer to an HIV test using oral fluid as an 'orientation test result'. In response, we now changed to wordings to ‘HIV diagnostic orientation test’ and the sentences in lines 487-488 and 709-711, respectively, now read as follows: “An HIV diagnostic orientation test will be used to assess the presence of antibodies in an oral fluid sample as described by Platteau et al. [43] and Loos et al. [37].” and “Up until now, large well-designed studies investigating HIV prevalence and associated risk factors in this heterogeneous population based on HIV diagnostic orientation testing are completely lacking.”

2. In lines 490-491, “pre-determined time sloths” should read “pre-determined time slots”

This comment is readily taken care of (see also minor comment 5 of reviewer 1).

---

## [Editor Report · Decision Letter 1]

15 Mar 2022

Prevalence and associated risk factors of HIV infections in a representative transgender and non-binary population in Flanders and Brussels (Belgium): Protocol for a community-based, cross-sectional study using time-location sampling

PONE-D-21-25105R1

Dear Dr. De Brier,

We’re pleased to inform you that your manuscript has been judged scientifically suitable for publication and will be formally accepted for publication once it meets all outstanding technical requirements.

Kind regards,

Hong-Van Tieu

Academic Editor

PLOS ONE
---

## [Editor Report · Acceptance letter]

1 Apr 2022

PONE-D-21-25105R1 

Prevalence and associated risk factors of HIV infections in a representative transgender and non-binary population in Flanders and Brussels (Belgium): Protocol for a community-based, cross-sectional study using time-location sampling 

Dear Dr. De Brier:

I'm pleased to inform you that your manuscript has been deemed suitable for publication in PLOS ONE. Congratulations! Your manuscript is now with our production department. 

Kind regards, 

on behalf of

Dr. Hong-Van Tieu 

Academic Editor

PLOS ONE